# Cerebellar Abscess Secondary to Cholesteatomatous Otomastoiditis—An Old Enemy in New Times

**DOI:** 10.3390/diagnostics13233566

**Published:** 2023-11-29

**Authors:** Andrei Ionut Cucu, Raluca Elena Patrascu, Mihaela Cosman, Claudia Florida Costea, Patricia Vonica, Laurentiu Andrei Blaj, Vlad Hartie, Ana Cristina Istrate, Iulian Prutianu, Otilia Boisteanu, Emilia Patrascanu, Adriana Hristea

**Affiliations:** 1Faculty of Medicine and Biological Sciences, University Stefan cel Mare of Suceava, 720229 Suceava, Romania; 2Emergency Clinical Hospital Prof. Dr. Nicolae Oblu, 700309 Iași, Romania; claudia.costea@umfiasi.ro (C.F.C.); laurentiu-andrei.blaj@d.umfiasi.ro (L.A.B.); liviu.vlad.hartie@umfiasi.ro (V.H.); istrate.ana-cristina@d.umfiasi.ro (A.C.I.); 3National Institute for Infectious Diseases Prof. Dr. Matei Bals, 021105 Bucharest, Romania; adriana.hristea@umfcd.ro; 4Infectious Diseases Department, Faculty of Medicine, University of Medicine and Pharmacy Carol Davila, 050474 Bucharest, Romania; 5Emergency County Hospital Braila, 810303 Braila, Romania; mihaelacosman@yahoo.com; 6Department of Ophthalmology, University of Medicine and Pharmacy Grigore T. Popa Iași, 700115 Iași, Romania; 7Department of Otorhinolaryngology, University of Medicine and Pharmacy Grigore T. Popa Iași, 700115 Iași, Romania; sonia.vonica@umfiasi.ro; 8Sf. Spiridon County Clinical Emergency Hospital Iași, 700111 Iași, Romania; otilia.boisteanu@umfiasi.ro; 9Department of Neurosurgery, University of Medicine and Pharmacy Grigore T. Popa Iași, 700115 Iași, Romania; 10Department of Anesthesia and Intensive Care, Grigore T. Popa University of Medicine and Pharmacy Iași, 700115 Iași, Romania; patrascanu.emilia@umfiasi.ro; 11Department of Morpho-Functional Sciences I—Histology, Faculty of Medicine, Grigore T. Popa University of Medicine and Pharmacy Iași, 700115 Iași, Romania; iulian-v-prutianu@d.umfiasi.ro; 12Regional Institute of Oncology, 700483 Iași, Romania

**Keywords:** otogenic cerebellar abscess, cholesteatoma, otomastoiditis, otitis, case report

## Abstract

Chronic otitis with cholesteatoma is a potentially dangerous disease that can lead to the development of intracranial abscesses. Although cerebellar abscess is half as common as cerebral abscess, it is known for its particularly difficult diagnosis, which requires the visualization of the pathological process continuity from the mastoid to the posterior fossa. In this article, we present an extremely rare case from the literature of cholesteatomatous otomastoiditis complicated with meningitis and cerebellar abscess, along with the description of technical surgical details for the plugging of the bony defect between the mastoid and posterior fossa with muscle and surgical glue. The particularity of this case lies in the late presentation to the doctor of an immunocompetent patient, through a dramatic symptomatology of life-threatening complications. We emphasize the importance of responsibly treating any episode of middle ear infection and considering the existence of underlying pathologies. In such cases, we recommend additional neuroimaging explorations, which can prevent potentially lethal complications. The treatment of such intracranial complications must be carried out promptly and requires collaboration between a neurosurgeon and an ENT surgeon.

## 1. Introduction

Chronic suppurative otitis media (CSOM) is a chronic inflammation of the middle ear associated with a perforated tympanic membrane and persistent purulent discharge [1] that has lasted for at least 6 weeks [2]. Epidemiological studies have shown varying percentages of CSOM depending on economic conditions, with the prevalence of this condition ranging from 0.3 to 14% [3]. For example, in the United States, the prevalence is less than 1% [2], while in India, it is approximately 7.8% [4]. Furthermore, the World Health Organization has established that a CSOM prevalence greater than 4% represents a critical public health issue for the affected population [5,6]. However, recent studies have indicated a decline in the incidence of CSOM and its complications over the past 15 years [7,8,9,10].

Cholesteatoma is a common condition found worldwide, especially in developing countries. Literally, it means “skin in the wrong place” and can be located in the external auditory canal, middle ear cavity, or in any pneumatized part of the temporal bone [11].

Regarding the incidence of cholesteatoma in suppurative middle ear infections, in a study that reviewed 12,000 patients with chronic middle ear suppuration, Nager identified the presence of cholesteatoma in one-third of the cases [12]. Sade et al. reported the existence of chronic middle ear infections in a community with a prevalence ranging from 0.5% to 30%. Furthermore, they estimate that approximately 20 million people worldwide are affected by chronic middle ear infections, and of these, around a quarter (about 5 million) have cholesteatomas [13,14].

As for the etiology of bone destruction caused by cholesteatomas and the creation of communications with the intracranial space, this is still a subject of debate. One theory suggests that the production resulting from enzymatic degradation by osteoclasts is the cause of bone destruction [15,16]. Therefore, due to its tendency to erode bone, cholesteatoma can lead to extremely severe and life-threatening intracranial complications [11]. The most significant intracranial complications of cholesteatomatous otomastoiditis include intracranial abscesses and meningitis, with varying rates in the literature [10].

Cholesteatomatous otomastoiditis can also spread into the intracranial space, with the cerebellum and temporal lobe being the most frequent locations for brain abscesses due to middle ear and mastoid infections [17]. Generally, cerebellar abscesses rank second after temporal lobe abscesses [18], yet cerebellar abscesses are known for the serious issues they present [19]. Due to the relatively small size of the posterior fossa, the effects of a cerebellar abscess and disproportionate edema in relation to the lesion’s size can be catastrophic [20].

Regarding the historical treatment of otogenic cerebellar abscesses, the Scottish surgeon William Macewan (1848–1924) [21] attempted the eradication of infection in the brain and ear [22]. He published the resolved cases in *Pyogenic Disease of the Brain and Spinal Cord: Meningitis, Abscesses of the Brain, Infective Sinus Thrombosis* in 1893. In several chapters, he addressed middle ear infections and their complications, such as mastoiditis, leptomeningitis, or cerebellar abscesses [23]. In 1898, in the treatment of cerebellar abscesses, Macewan reported a mortality rate of 55% in nine operated cases, thus demonstrating the lethal nature of this condition and the importance of pus drainage [24,25]. Subsequently, in 1973, Samson and Clark, in a review of cerebral abscesses, concluded that the high mortality in such cases was primarily due to increased intracranial pressure rather than the infection itself. Therefore, they recommended reducing the mass effect of the abscess through its total excision and antibiotic therapy [26]. In a work on cerebellar abscesses, Pennybacker reported higher survival rates in patients with cerebellar abscesses who, in addition to surgical decompressive procedures, also received antibiotics [27].

In the past, studies reported that 25% of all brain abscesses in children were otogenic, and in adults, over 50% of them had otogenic origins [28,29]. The development of antibiotics and the availability of advanced imaging techniques have reduced these incidence rates and mortality, especially in developed countries [30].

In this case report, we present the case of a 40-year-old man who was admitted to our department with cholesteatomatous otomastoiditis complicated by a cerebellar abscess. The uniqueness of this case lies in the late presentation to the doctor of an immunocompetent patient, through a dramatic symptomatology caused by these two life-threatening complications: meningitis and cerebellar abscess. However, the condition was ultimately successfully resolved through a multidisciplinary team.

## 2. Case Report

### 2.1. Patient Information and Clinical Findings

The subject of this case-report is a 40-year-old male who presented to our neurosurgery department with a significant headache, fever, and cerebellar symptoms and signs (Table 1), which appeared insidiously six days before hospitalization. One day before admission, the aforementioned symptoms worsened, to which neck stiffness was added.

From the patient’s medical history, we found that he had left-sided otorrhea for 2 months, for which he was taking various antibiotics. He was treated on an outpatient basis for left-sided chronic middle ear infection by an ENT specialist for 2 weeks, and subsequently presented to the hospital due to worsening symptoms. Also, we found out that he had multiple drug allergies (to Vancomycin, Meronem, Cefort, and Gentamicin). On physical examination, the patient was conscious, with normal vital signs, but exhibited neurological deficits, including left peripheral facial palsy (House-Brackmann grade III) and cerebellar signs and symptoms (Table 1). Signs of meningeal irritation were positive, including neck stiffness and Kernig’s sign.

The otoscopic examination revealed medium amounts of purulent secretions discharged through the left external auditory canal. Additionally, it revealed a well-defined prolapsed lesion from the upper wall of the external auditory canal, beside which cerebrospinal fluid was being discharged. The tympanic membrane and the chain of ossicles were destroyed. Relevant physical examination findings are summarized in Table 1.

### 2.2. Diagnostic Assessment

#### 2.2.1. Laboratory Studies

Abnormal results were initially obtained in the emergency room, namely:Complete Blood Count (CBC)—leukocitosis of 10,300/µL (4000–10,000/µL)Neutrophilia—7250/µL (1100–7000/µL)Erytrocite sedimentation rate—79 (0–10/mm/1 h)Serum C-protein reactive (CRP)—4.8 (0–1.0/mg/dL)Fibrinogen—522 (150–400/mg/dL)HIV test—negativeBlood cultures in fever, cultures from the ear—negative.

#### 2.2.2. Imaging Studies

The computed tomography (CT) scan of the head revealed a left cholesteatomatous otomastoiditis (Figure 1A) and a lesion in the left cerebellar hemisphere, which appeared to have continuity with the cavity from the left mastoid (Figure 1B).

Furthermore, the contrast-enhanced CT scan revealed a left cholesteatomatous otomastoiditis (Figure 2).

The head CT scan was complemented by magnetic resonance imaging (MRI) with contrast, which revealed a well-defined encapsulated lesion in the left cerebellar hemisphere. The lesion had a heterogeneous structure (hypointense on T1, hyperintense on T2 and FLAIR), with significant diffusion restriction and enhancement on the periphery when contrast was administered. It was associated with perilesional edema and a mass effect that caused compression of the fourth ventricle (measuring 43/36/20 mm—AP/T/CC) as shown in Figure 3.

Furthermore, the head MRI scan also revealed soft tissues and inflammation in the left mastoid region, with a connection to the left cerebellar abscess, raising suspicion of a mastoid abscess (Figure 4).

### 2.3. Therapeutic Intervention

An emergency surgical intervention was performed, involving the evacuation of the cerebellar abscess through a left suboccipital craniectomy. A follow-up CT scan of the head showed successful evacuation of the abscess, but the communication between the mastoid and the intracranial space remained (Figure 5).

After obtaining biological material, intravenous antibiotic therapy was initiated with Ciprinol 100 mg/10 mL twice a day, Metronidazole 200 mg twice a day, and Clindamycin 300 mg three times a day throughout the patient’s hospitalization in our department (30 days). Cultures from the cerebellar abscess turned out to be negative.

ENT procedures were performed on the third day, once the patient’s condition had stabilized. The purpose of the surgery was to remove the cholesteatoma, the underlying cause of the complications. After creating a retroauricular incision, the mastoid was drilled (Figure 6). At this point, pus was observed coming from the exposed extradural space, and it was aspirated using suction. Subsequently, the antrum was uncovered, which was found to be occupied by a large, infected cholesteatoma, extending into the aditus, attic, and intersinuso-facial cells, and towards the apex of the mastoid. The cholesteatoma was carefully removed, and it necessitated a canal wall down procedure. Granulation tissue and the cholesteatoma were also removed from the mastoid, and purulent secretions were aspirated from the tympanic cavity. During the surgery, we observed that the wall of the facial canal was mildly eroded by the cholesteatoma.

In the end, we examined the cells and antrum, and removed all the lesions under the microscope until only healthy bone tissue remained. The dural and bony defect in the posterior fossa was then sealed with a free muscle graft harvested from the occipitalis muscle, and the adherence was achieved using surgical adhesive glue (Figure 7).

In the end, a large cavity remained, which we utilized for draining the remaining pus. Cultures of the pus from the mastoid process and cerebellar abscess were sterile, likely due to the patient’s prolonged antibiotic therapy. The patient was monitored with head CT scans at 2 and 6 months, and after 30 days of antibiotics, we observed the closure of the bony defect through which the cholesteatoma cavity communicated with the posterior fossa (Figure 8).

The patient was discharged from the hospital in good condition with dry ears, experiencing only mild facial paralysis and slight dizziness. The patient was transferred to an ENT department where intravenous antibiotic therapy was continued.

## 3. Discussion

In Western countries, otogenic brain abscess represents approximately 1–2% of space-occupying brain lesions, while in developing countries, this condition rises dramatically to 8% [31]. After meningitis, cerebellar abscess is the second most common complication caused by middle ear infections [31,32,33].

### 3.1. Anatomical Pathways for the Spread of Infection to the Intracranial Space

Otogenic intracranial abscesses are typically caused by the direct extension of an infection from the temporal bone into the intracranial cavity. This usually occurs through: (i) extension through bone that has demineralized during acute infection, or resorption by cholesteatoma or osteitis in chronic destructive disease. For example, osteitis may cause a bony defect in the tegmen tympani (for a temporal abscess) or in Trautmann’s triangle (for a cerebellar abscess), (ii) retrograde thrombophlebitis of the lateral sinus, which occurs by the spread of infected thrombi, (iii) through preformed anatomical pathways, such as the oval or round window, internal auditory canal, cochlear and vestibular aqueducts, congenital bony defects like the facial canal, or tegmen plate, or (iv) acquired bony defects resulting from fractures, neoplasms, stapedectomy, or implantable devices [34,35].

CSMO is a long-standing middle ear infection associated with the perforation of the tympanic membrane and persistent discharge [4]. The perforation is permanent due to the fact that the edges of the perforation are lined with squamous epithelium, which prevents the natural closure of the perforation. In these cases, external drainage lasts for more than 12 weeks [4]. It usually begins with irritation and inflammation of the middle ear mucosa, caused by aerobic bacteria [36]. This inflammatory response results in mucosal swelling and increased middle ear discharge, ultimately leading to tympanic membrane perforation [4]. These perforations are divided into two types: tubotympanic and atticoantral (Table 2). Atticoantral perforations typically always involve cholesteatoma and are a dangerous and unsafe type [37].

The atticoantral type is dangerous because it involves the postero-superior part of the cleft, including the attic, antrum, and mastoid. The risk of complications is very high, with the most dreaded complications being intracranial, such as subperiosteal abscess, meningitis, brain, or cerebellar abscess. In the case of our patient, the pathogenesis of the otogenic cerebellar abscess can be explained by direct erosion of bone into the posterior fossa.

### 3.2. Microbial Etiology

The microbial etiology of cerebral abscess depends on the patient’s age, the primary site of infection, and the patient’s immune status [38], and is often polymicrobial. The most frequently implicated pathogens are *Staphylococcus aureus*, *Streptococcus pneumoniae, Enterobacteriaceae* (*Proteus* spp., *Escherichia coli*, *Klebsiella pneumoniae*), *H. Influenzae*, *Pseudomonas* spp., and *Neisseria meningitidis*. With the increasing number of immunocompromised patients, including those with HIV/AIDS, transplant recipients, and those on immunosuppressive medication, there has been a rise in the incidence of brain abscesses caused by opportunistic pathogens [38].

### 3.3. Signs and Symptoms

Cerebral abscess typically presents with the following symptoms: headache in 82.5% of cases, altered level of consciousness in 75% of cases, and papilledema in 66.9% of cases [33]. Neurological symptoms depend on the abscess’s location, with cerebellar abscesses possibly manifesting as cranial nerve palsies, gait disturbances, headaches, or altered consciousness due to hydrocephalus [24]. Our patient presented with severe occipital headache, vomiting, fever, signs of meningeal irritation, cerebellar signs, peripheral paralysis of the left facial nerve, and purulent discharge from the left ear. Clinical signs of hydrocephalus were not present because the cerebellar abscess was located in the cerebellar hemisphere, causing minimal compression of the fourth ventricle. In a systematic review, Duarte et al. suggested that in the presence of otogenic infection with a “red flag” such as fever, headache, nausea, vomiting, altered mental status, or any localizing neurologic signs, one must accelerate the laboratory tests and cultures and additional imaging studies should be performed [39].

### 3.4. Laboratory Studies

Common laboratory tests indicated in cases of suspected cerebellar abscess include: (i) CBC, erythrocyte sedimentation rate, CRP, (ii) blood cultures if there is a fever, cultures from ear secretions (recommended at least twice, preferably before starting antibiotic therapy), and (iii) culture of the cerebellar abscess for aerobic and anaerobic bacteria, Gram staining, acid-fast staining, and special fungal stains [38]. Lumbar puncture is contraindicated in the case of large intracranial lesions due to increased intracranial pressure, which can lead to brain herniation and death. Lumbar puncture is recommended only after imaging exploration (CT or MRI scanning) and if those permit it.

### 3.5. Imaging Studies

Radiographic imaging studies (such as mastoid Schuller, Chausee III, and Stenvers views) are rarely used today and only demonstrate bone erosion and an abnormal position of the lateral sinus or tegmen tympani. In cases of suspected cerebellar abscess, all patients should undergo a cranial CT scan [40,41]. A head CT scan with contrast enhancement is the method of choice for diagnosing cerebellar abscess and associated mastoiditis [42,43]. Additionally, CT scans with contrast are used to monitor progression after treatment. CT imaging is the primary investigation capable of examining the lesion in the ear, mastoid, the extension of pus, the condition of the semicircular canals, the facial nerve, internal or external aspects of the mastoid bone, as well as cerebral edema, pus collection, thrombosis, encephalitis, or hydrocephalus [38].

Britt et al. described three stages of cerebral abscess evolution that can be observed in computerized tomographic imaging: stage I (early cerebritis stage, days 1–3), stage II (late cerebritis stage, days 4–9), and stage III (capsule formation stage, days 10–14) [44,45]. In the case of our patient, the abscess was in stage III, the stage of capsule formation, and in such cases, once the abscess capsule is formed, surgical intervention is essential [46,47,48,49].

### 3.6. Medical Treatment

Broad-spectrum antibiotics that can cross the blood-brain barrier in sufficient concentrations are the first line of treatment. Delay in initiating antibiotic therapy can lead to a negative prognosis [50]. The duration of antibiotic treatment depends on the patient’s clinical course and follow-up imaging with an MRI or CT scan. However, in patients with bacterial brain abscess, traditional intravenous antibiotic therapy should typically last for 6–8 weeks.

Among the recommended antibiotics are Vancomycin, which is effective against MRSA and *S. epidermidis*, aerobic/anaerobic streptococci, and *Clostridium*. Metronidazole is recommended for its ability to penetrate the central nervous system and is used in infections with anaerobic bacteria. Third-generation cephalosporins (ceftriaxone, cefotaxime) provide protection against aerobic gram-negative pathogens and aerobic streptococci. Additionally, fourth-generation cephalosporins (cefepime, ceftazidime) are recommended for infections with Pseudomonas. Although they penetrate the CNS, fluoroquinolones have limited use in the treatment of cerebral abscesses. Meropenem is used in high doses to increase the chances of survival. Injecting antibiotics into the residual cavity of the abscess is not necessary because many antibiotics penetrate the brain.

In the medical treatment of cerebral or cerebellar abscess, other medications may also be used. Cerebral edema may require urgent depletive treatment, which can be achieved with mannitol or Dexamethasone. Mannitol can be administered as a 20% solution at a dose of 0.25–0.5 g/kg every 3–5 h, with monitoring of osmolarity and serum electrolytes. Dexamethasone is effective in reducing cerebral edema and is typically given intravenously at a dose of 4 mg every 4–6 h when signs of intracranial hypertension are present. Additionally, anticonvulsants are recommended in the case of cerebral abscesses.

### 3.7. Surgical Treatment

The management of most cerebral abscesses involves a combination of surgical and antibiotic treatment. The preferred surgical treatment is neurosurgical drainage, especially when intracranial abscesses have a diameter greater than 2.5 cm [50]. In the case of cerebellar abscesses, in the past, Griffith recommended extensive decompression of the posterior fossa [51], particularly when patients deteriorated rapidly from a normal conscious state to deep coma within a few minutes [20].

Referring to the three stages of cerebral abscess evolution described by Britt et al., surgical drainage of the abscess is recommended for mature abscesses, especially those of larger dimensions, which have not responded to antibiotic therapy or when there is a neurological deficit. While antibiotic therapy is effective in the early and late stages of cerebritis [43,52,53], in the stage of capsule formation, antibiotics alone are ineffective due to the acidic medium in the abscess cavity and their inability to reach adequate therapeutic concentrations within the abscess [54]. In such cases, surgical intervention is vital, both for the removal of the abscess capsule and the abscess itself [52,55,56]. After abscess surgery, early management of the ear disease is recommended, such as a radical mastoidectomy for source control [39], and antibiotic therapy for 6–8 weeks [57,58,59,60]. Otologic procedures to clear the source of infection are mandatory. Once the patient becomes stable, a radical mastoidectomy and canal wall-down procedure are recommended.

The timing of ear surgery and cerebellum surgery varies from author to author [61]. In the past, it was considered that neurosurgery should precede ear surgery because mastoid surgery alone could not prevent the development of the abscess [62]. In a study that included all patients treated for otogenic intracranial abscesses between 1970 and 2012 at a tertiary center in Finland, the authors reported that 69% of interventions for intracranial abscesses were performed before ear surgery [63]. Additionally, Bradley et al. mentioned that in most cases, the evacuation of the cerebral abscess was performed first, followed by mastoidectomy at a second operation [64].

In a more recent study from 2013 conducted in China, Sun et al. reported that cerebral abscess drainage and mastoidectomy were performed simultaneously, with all patients undergoing these procedures within 1 day of admission [65]. Nathoo et al. also reported that in all cases, mastoidectomy was performed at the same time as the neurosurgical intervention [66]. Furthermore, studies comparing patients operated on in the same session with those operated on in two separate surgical sessions have concluded that there is no difference in outcomes [67]. In other words, coordination between otology and neurosurgical teams, whether in a single-stage or delayed fashion, is key in the treatment of otogenic abscesses where source control must be obtained through mastoidectomy [39]. However, in cases where the patient experiences neurological deterioration, the cerebellar abscess should be evacuated first by a neurosurgical team, followed by mastoidectomy for source control, as was done in the case of the patient in question.

Regarding the closure of the defect between the mastoid and the posterior fossa or middle fossa, this can be achieved with bone plates [68,69], bone chips [70], muscle [71,72], or fat [73]. Among these materials, bony material is harder to resorb than soft tissue [73]. Since the bone defect in our patient was of considerable size, and yet it closed, we consider the technique of sealing communication with muscle and surgical adhesive glue to be effective and valuable to be taken into account in such cases.

Cases of cholesteatomatous otomastoiditis complicated with cerebellar abscess are rare in developed countries and are usually encountered in individuals with poor hygiene or low medical education. Such cases can be immediately life-threatening if not promptly treated, so any danger signs related to long-standing otitis media should be investigated with cranial and cerebral imaging to prevent deadly complications [74].

## 4. Conclusions

The development of a cerebellar abscess after cholesteatomatous otomastoiditis is extremely rare and can be considered a historical condition. In this regard, very few cases have been documented in the literature up to the present day. Our case describes a young male patient who had been suffering from cholesteatomatous otomastoiditis and presented with a cerebellar abscess and meningitis.

The main take-home message from this case is that any episode of middle ear infection should be treated responsibly, considering the possibility of an underlying condition, especially when otologic symptoms persist. Imaging studies such as CT or MRI scans are crucial for early diagnosis, not only of complications but also of the primary otogenic focus. On the other hand, the coordination and collaboration between the neurosurgeon and ENT surgeon are essential for the prompt and successful resolution of such cases.

## Figures and Tables

**Figure 1 diagnostics-13-03566-f001:**
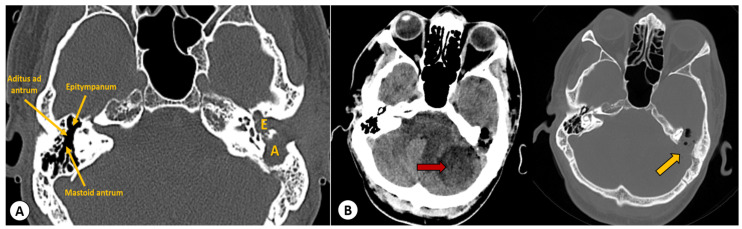
(**A**): Axial CT scan showing a left cholesteatomatous otomastoiditis. In the left ear, you can observe widening of the aditus [A] and the formation of a common cavity between the epitympanum [E] and the aditus with soft tissue within. In the right ear, normal anatomical structures can be seen: the epitympanum, aditus ad antrum, and mastoid antrum. (**B**) Axial head CT scan showing a well-defined hypodense lesion located in the left cerebellar hemisphere (red arrow). Moreover, in the bone window, erosion of the bony wall of the temporal bone can be observed, indicating a continuity between the left cholesteatomatous otomastoiditis and the posterior fossa (yellow arrow).

**Figure 2 diagnostics-13-03566-f002:**
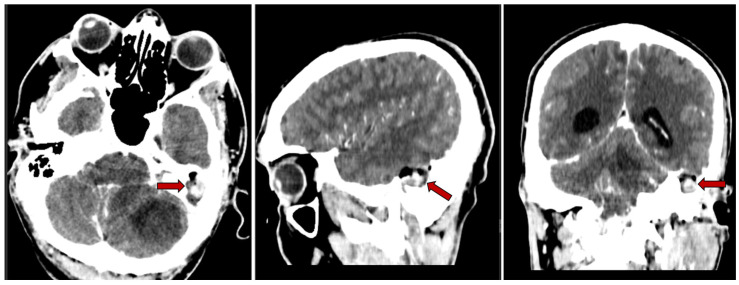
Contrast-enhanced CT scan (axial, sagittal and coronal planes) showing a left cholesteatomatous otomastoiditis (red arrow).

**Figure 3 diagnostics-13-03566-f003:**
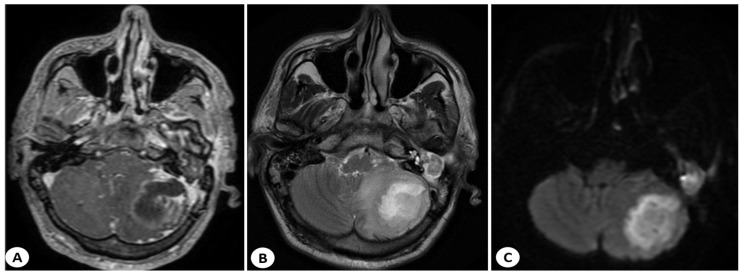
Axial MRI scan showing a cerebellar abscess in the left hemisphere. In the T1-weighted contrast-enhanced axial section (**A**), a round mass with peripheral contrast enhancement can be seen. In the T2-weighted axial image (**B**), its heterogeneous structure is visible, and in the DWI sequence (**C**), diffusion restriction can be observed; this round mass with peripheral contrast enhancement is in contact with a soft tissue mass from the left middle ear.

**Figure 4 diagnostics-13-03566-f004:**
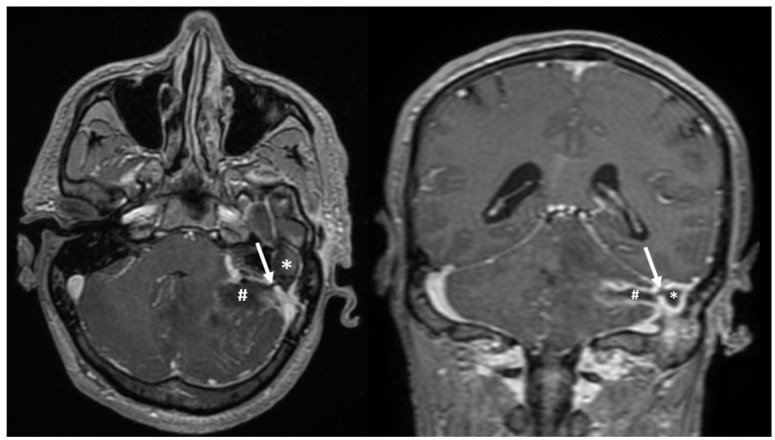
Axial and coronal MRI scan showing the presence of a soft tissue mass in the left mastoid (*), and communication (white arrow) with a cerebellar abscess in the left hemisphere (#); peripheral contrast enhancement can be observed in both lesions.

**Figure 5 diagnostics-13-03566-f005:**
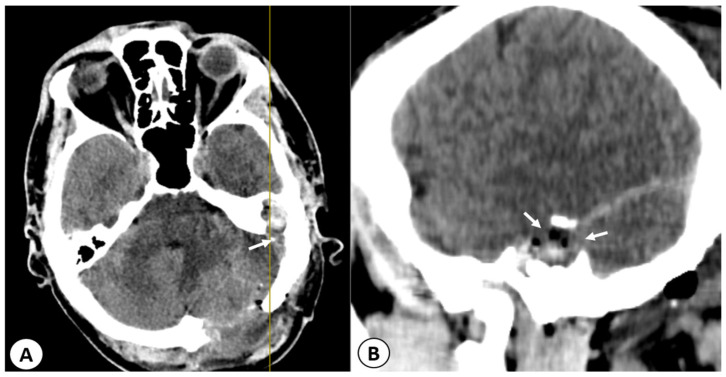
(**A**) Postoperative CT scan reveals the evacuation of the cerebellar abscess from the posterior fossa but with the retention of communication (white arrow) between the cholesteatoma cavity in the left mastoid and the posterior fossa. (**B**) Postoperative CT scan showing the communication of the cholesteatoma through the tegmen tympani with the middle fossa, and posteriorly with the posterior fossa (white arrows).

**Figure 6 diagnostics-13-03566-f006:**
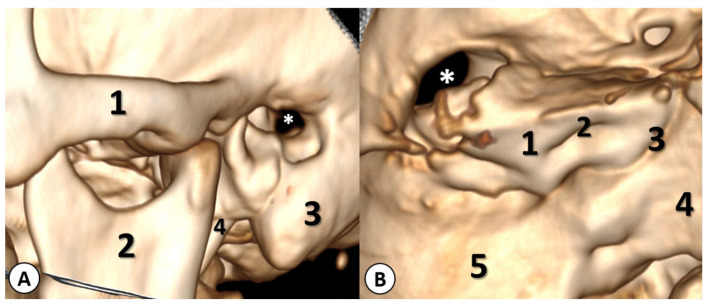
Head CT imaging 3D reconstructions highlighting the left antrostomy: (**A**) External view of the left auricular region, showing the communication made with the cholesteatoma cavity (white asterisk): 1—zygomatic arch, 2—ramus of mandible, 3—mastoid process, 4—styloid process. (**B**) Skull base with 1—petrous bone, 2—internal auditory meatus, 3—petroclival fissure, 4—clivus, 5—posterior fossa. White asterisk represents the communication made with the cholesteatoma cavity.

**Figure 7 diagnostics-13-03566-f007:**
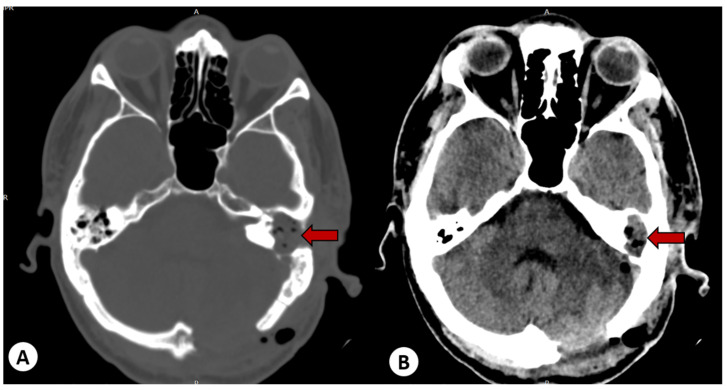
Head CT scan (**A**—bone window), (**B**—brain window) showing the presence of the muscle in the cholesteatoma cavity (red arrow), which seals the communication with the posterior fossa and middle fossa.

**Figure 8 diagnostics-13-03566-f008:**
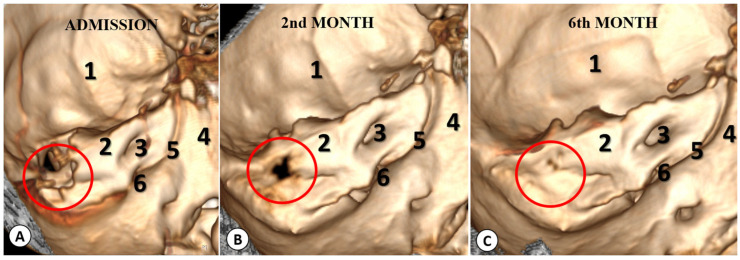
Head CT imaging 3D reconstructions at admission (**A**), 2 months (**B**), and 6 months (**C**) showing the dynamic closure of the bony defect between the cholesteatoma cavity and the posterior fossa (red circle). The identified anatomical elements are: 1—middle floor, 2—petrous bone, 3—internal auditory meatus, 4—clivus, 5—petroclival fissure, and 6—jugular foramen.

**Table 1 diagnostics-13-03566-t001:** Summary of physical examination.

Targeted System	Findings
General physical examination	Blood pressure—120/70 mmHg Temperature—37 °C Pulse rate—100/min Respiratory rate—18/min SpO_2_—97% at room air Yellow colored discharge present in the left external auditory canal
2. Central nervous system	GCS (Glasgow Coma Scale)—15 points Drowsiness Neck stiffness and Kernig’s sign positive Left peripheral facial palsy (House-Brackmann grade III) Cerebellar signs and symptoms: ○ cerebellar ataxia ○ dysmetria with hypermetria ○ adiadochokinesia ○ asynergy ○ balance disorders with positive Romberg test ○ horizontal nystagmus
3. Gastrointestinal and cardiovascular systems	No significant findings related to case

**Table 2 diagnostics-13-03566-t002:** Differences between tubotympanic and attico-antral types of CSOM.

	Tubotympanic (Safe Type)	Atticoantral (Unsafe Type)
Discharge	Profuse, mucoid	Scanty, purulent, foul-smelling
Perforation	Central	Attic or marginal
Granulations	Uncommon	Common
Cholesteatoma	Absent	Present
Complications	Rare	Common

## Data Availability

All data are reported in the text.

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
