# Peer review of "Cerebellar Abscess Secondary to Cholesteatomatous Otomastoiditis—An Old Enemy in New Times"

_diagnostics, 2023, doi:10.3390/diagnostics13233566_

Round 1
Reviewer 1 Report
Comments and Suggestions for Authors
The paper is very well written and can be accepted for publication in the current form.
Author Response
Reviewer 1
The paper is very well written and can be accepted for publication in the current form.
The authors would like to thank reviewer 1 for their words of appreciation. We thank you and are grateful.

Reviewer 2 Report
Comments and Suggestions for Authors
The current manuscript presented a patient with a cerebellar abscess due to cholesteatoma, and his therapeutic approach was described. The manuscript is well-written but there are some issues and concerns about it:
- There are several manuscripts in the literature that reported the otogenic cerebellar abscess, presentations, and treatments. What makes your manuscript distinct from these reports? What does your manuscript add to our present knowledge?
- The chief complaint of the patient needs more details, such as when the symptoms begin.
- The past medical history of the patient is not complete. How long did he suffer from otorrhea? Has he ever been treated for otitis media or mastoiditis? when and how?
- In physical examination, the otoscopic findings need to be completed
- Balance disturbances need more details
- What were the audiometric findings of the patient preoperatively and postoperatively?
- What were the features of facial nerve palsy? was the palsy complete? when did the palsy happen? What were the surgical findings concerning facial nerve pathology?
- What were your intraoperative findings regarding the ossicular chain? did you perform tympanoplasty or ossiculoplasty?....
- Figure 1(B) shows the vicinity of the cholesteatoma with abscess, but not continuity
Author Response
Reviewer 2
The current manuscript presented a patient with a cerebellar abscess due to cholesteatoma, and his therapeutic approach was described. The manuscript is well-written but there are some issues and concerns about it:
Thank you for the words of appreciation related to the writing of the manuscript. Regarding the aspects identified in the manuscript, we will answer you in detail in the following
- There are several manuscripts in the literature that reported the otogenic cerebellar abscess, presentations, and treatments. What makes your manuscript distinct from these reports? What does your manuscript add to our present knowledge?
Indeed, the literature reports several cases of otogenic cerebellar abscess, especially during the years 1990-2000. Very few cases are reported after the year 2000, precisely because this complication (intracranial abscess) of otomastoiditis is considered historical in our times, when imaging methods are so advanced. Our manuscript brings to light a rare complication of cholesteatomatous otomastoiditis, namely the cerebellar abscess, in the conditions where most of the complications are represented by the temporal abscess.
We consider that this case report is distinct from other similar case reports due to:
1 - The article presents an immunocompetent patient who presented very late to the doctor, leading to the emergence of dramatic and life-threatening complications (meningitis and cerebellar abscess).
2 - The bony defect was very significant, and we report that in such cases, consideration can be given to filling with muscle and surgical glue, a technique that proved to be effective in our case.
3 - We raise an alarm signal that in cases of episodes of otitis media, this underlying pathological condition (cholesteatom) should also be considered, especially when symptoms persist, and treatment seems ineffective.
- The chief complaint of the patient needs more details, such as when the symptoms begin.
We have added details about the onset of symptoms in the text.
- The past medical history of the patient is not complete. How long did he suffer from otorrhea? Has he ever been treated for otitis media or mastoiditis? when and how?
We have added details in the text.
- In physical examination, the otoscopic findings need to be completed
We have added details in the text.
- Balance disturbances need more details
We have added details in the text.
- What were the audiometric findings of the patient preoperatively and postoperatively?
Audiometry was not necessary as the patient had complete deafness in the left ear due to the total destruction of the eardrum and the ossicular chain.
- What were the features of facial nerve palsy? was the palsy complete? when did the palsy happen? What were the surgical findings concerning facial nerve pathology?
The left facial nerve paralysis was classified as Grade II (House-Brackmann scale), present upon admission, and did not change postoperatively. Intraoperatively, the wall of the facial canal was mildly eroded by cholesteatoma. The impairment of the left facial nerve remained constant.
- What were your intraoperative findings regarding the ossicular chain? did you perform tympanoplasty or ossiculoplasty?....
Both the eardrum and the ossicular chain were completely destroyed. Neither tympanoplasty nor ossiculoplasty were performed.
- Figure 1(B) shows the vicinity of the cholesteatoma with abscess, but not continuity
The CT section in figure 1 (B) did not pass through the area of communication between the cholesteatoma and the cerebellar abscess, which was better highlighted in the contrast-enhanced MRI sequences (Figure 3). To improve the imaging comprehension, following valuable recommendations, we have modified figure 1 (B) with another CT sequence that clearly highlights the communication between the cholesteatoma and the cerebellar abscess.

Reviewer 3 Report
Comments and Suggestions for Authors
The article is well written. Even if this is a well known condition, I think it should be published.
Author Response
Thank you for the words of appreciation. Indeed, we also believed that publishing this case was useful, firstly because it was successfully resolved, and secondly, because it highlights a classic complication of cholesteatomatous otitis, which, as it appears, is not entirely a thing of the history. Additionally, the patient's delayed presentation to the doctor with life-threatening late complications is an element of interest. Thank you once again for the words of appreciation

Reviewer 4 Report
Comments and Suggestions for Authors
This is a case report aimed at elaborating on the diagnostic and treatment details of otogenic cerebellar abscess.
The abstract is adequate, and has listed rationale and setting details, alongside the most important findings, but I would urge the authors to reorganize the title, abstract and paper subheadings according to CARE guidelines.
The objectives of the study are presented clearly and the introduction section communicates the need for systematic reporting of otogenic cerebellar abscess as a rare but lethal complication of CSOM.
Again, the manuscript sections should adhere to the CARE guidelines, available on the Care website (https://www.care-statement.org/) alongside a CARE checklist (Introduction, Case Report, Discussion and Conclusion). Informed consent and IRB approval or waiver of thereof should be stated in the Case report section.
I especially enjoyed the introduction section and the historical details presented. The case report section shoul include comments on the course of the facial nerve, since a preoperative palsy was reported, and should include a House Brackmann grading score. Comments on possible obliteration extent and techniques are also welcome – apart from muscle grafting, did the authors consider other techniques to close the mastoid?
The discussion is very comprehensive, detailed and interesting. I would suggest stressing the novelty and learning points in this case report – the comprehensive discussion and literature review, since the condition itself is historic (although not so rare in the post-covid setting, as the reviewer has already had four patients like this in the past year and-a-half, and can appreciate the author’s trouble).
Author Response
This is a case report aimed at elaborating on the diagnostic and treatment details of otogenic cerebellar abscess.
The abstract is adequate, and has listed rationale and setting details, alongside the most important findings, but I would urge the authors to reorganize the title, abstract and paper subheadings according to CARE guidelines.
Unfortunately, the options for the title are limited, because the title itself is very long and already contains the keywords. One option we suggest is:
A case of cerebellar abscess secondary to cholesteatomatous otomastoiditis successfully resolved. Case report
Considering that the title, according to the recommendations of CARE, should contain the diagnosis or the primary intervention, we opt for the diagnosis of the case, since the surgical intervention is less relevant from a scientific point of view (evacuation of a cerebellar abscess and sealing of an acquired bone defect).
The objectives of the study are presented clearly and the introduction section communicates the need for systematic reporting of otogenic cerebellar abscess as a rare but lethal complication of CSOM.
The authors wish to express their thanks and gratitude for the reviewer's words of appreciation.
Again, the manuscript sections should adhere to the CARE guidelines, available on the Care website (https://www.care-statement.org/) alongside a CARE checklist (Introduction, Case Report, Discussion and Conclusion). Informed consent and IRB approval or waiver of thereof should be stated in the Case report section.
We restructured the manuscript according to CARE recommendations.
We have added more detailed and coherent information regarding the patient's symptoms, their onset, evolution and relevant medical interventions. We also added information about the otoscopic and clinical examination of the patient.
The authors declare that they obtained the patient's consent for the publication of this case. It was also requested by the Editorial board of the journal and a scanned copy of this consent was sent. Also, the information about the patient's consent was mentioned at the end of the manuscript. Since it is a case report, Institutional review board statement is not applicable in this case.
I especially enjoyed the introduction section and the historical details presented. The case report section shoul include comments on the course of the facial nerve, since a preoperative palsy was reported, and should include a House Brackmann grading score. Comments on possible obliteration extent and techniques are also welcome – apart from muscle grafting, did the authors consider other techniques to close the mastoid?
Thank you for the words of appreciation. According to the recommendations of reviewer 2, we have already added the House-Brackman grading score. It was an important an necessary addition, thank you for the valuable recommendation.
Intraoperatively, the wall of the facial canal was discretely eroded by the cholesteatoma. This information was also added to the text.
Regarding the filling techniques of the acquired bone defect, unfortunately, in the literature we have identified only a few sources that mention filling with muscle (sources 71 and 72), bone fragments (sources 68-70) or fat (source 73). We did not use other techniques to close the mastoid in this case, as obliteration of the communication was successfully achieved by using muscles and adhesive glue.
The discussion is very comprehensive, detailed and interesting. I would suggest stressing the novelty and learning points in this case report – the comprehensive discussion and literature review, since the condition itself is historic (although not so rare in the post-covid setting, as the reviewer has already had four patients like this in the past year and-a-half, and can appreciate the author’s trouble).
Thank you for the words of appreciation; we have added new discussion points and conclusions for learning. Although initially we was very skeptical about closing the bone defect with just a piece of muscle, it seems to have worked, with CT images in dynamic mode showing ossification closure of this bone defect between the mastoid and the posterior fossa. Thank you for sharing thoughts; indeed, it was a challenging case, considering the patient's multiple antibiotic allergies, which reduced our crucial antibiotic arsenal (Vancomycin, Meronem, Cefort, etc.).

Round 2
Reviewer 2 Report
Comments and Suggestions for Authors
I would like to thank the authors for addressing all the issues.